# Insight on Mercapto-Coumarins: Synthesis and Reactivity

**DOI:** 10.3390/molecules27072150

**Published:** 2022-03-26

**Authors:** Eslam Reda El-Sawy, Ahmed Bakr Abdelwahab, Gilbert Kirsch

**Affiliations:** 1National Research Centre, Chemistry of Natural Compounds Department, Dokki, Cairo 12622, Egypt; 2Plant Advanced Technologies (PAT), 54500 Vandœuvre-lès-Nancy, France; ahm@plantadvanced.com; 3Laboratoire Lorrain de Chimie Moleculaire (L.2.C.M.), Universite de Lorraine, 57050 Metz, France

**Keywords:** mercapto-coumarins, diazosulfuration, Newman–Kwart, fluorescence, biological activity

## Abstract

Mercapto (or sulfanyl)-coumarins are heterocycles of great interest in the development of valuable active structures in material and biological domains. They represent a highly exploitable class of compounds that open many possibilities for further chemical transformations. The present review aims to draw focus toward the synthetic applicability of various forms of mercapto-coumarins and their representations in pharmaceuticals and industries. This work covers the literature issued from 1970 to 2021.

## 1. Introduction

Coumarins (2*H*-1-benzopyran-2-ones) are an elite class of compounds present in various natural products, and they have wide applications, *viz*., as additives in food [1,2], perfumes [3], cosmetics [4], and pharmaceuticals [5,6], as well as in the preparation of optical brighteners [7], dispersed fluorescent [8,9,10] and laser dyes [11], and useful medicinal products [12,13]. On the other hand, the carbon–sulfur bond formation plays an important role in organic synthesis [14,15,16,17]. The introduction of the thiol group to organic structures has emerged as an important tool in medicinal chemistry and chemical biology [18,19,20,21]. It plays a distinguished role in the fabrication of applicable substances in the field of advanced functional materials [22], structural frameworks of natural products [23], and the pharmaceutical industry [24,25,26]. Therefore, there is an increasing demand to investigate thiol-based coupling reactions focusing on their chemoselectivity and their tolerance of various functional groups in order to provide feasible access to new chemical architectures [27,28].

The incorporation of a thiol functional group into coumarin results in mercapto-coumarins. Although mercapto-coumarins have been relatively less extensively studied [18,19,20,21], their chemistry and bioactivity appear to be interesting. This functionalization of coumarin allows a special reactivity due to the implication of the thiol group in different types of organic reactions. This facilitates access to various series of derivatives that may have special applications or biological activities.

By exploring mercapto-coumarin derivatives, we found that four common forms of functional thiol group integrate into the coumarin moiety that occupies different positions, either on the pyrone ring or on the benzene ring. The common four mercapto-coumarins are 3-mercapto-coumarin, 4-mercapto-coumarin, 6-mercapto-coumarin, and 7-mercapto-coumarin (Figure 1).

The present review is concerned with the period from 1970 to 2021 to shed light on the different pathways of mercapto-coumarin synthesis, while also covering their broad applications at both industrial and biological levels. In addition, it is a groundbreaking release on these compounds, to open a platform for researchers to progress the development of this chemistry.

## 2. 3-Mercapto-Coumarin

By analyzing the synthesis of 3-mercapto-coumarin, we found that the source of sulfur was a heterocyclic compound not an inorganic reagent. In addition, the coumarin was formed in situ from primary sources, which were salicylaldehydes.

Qiyi et al. reported the synthesis of 3-mercapto-coumarin (**4**) from 2-hydroxybenzylidenerhodanine (**3**). The latter was produced in situ from salicylaldehyde (**1**) and 2-thioxothiazolidin-4-one (**2**). The reaction proceeded to the final target by refluxing of compound **3** in diluted ethanolic sodium hydroxide solution (Figure 1) [18].

In 2009, a green catalyst-free synthetic protocol for synthesizing varieties of the target 3-mercapto-coumarins was reported. In this protocol, refluxing of 2-methyl-2-phenyl-1,3-oxa-thiolan-5-one (**5**) and salicylaldehyde derivatives in water afforded the formation of corresponding 3-mercapto-coumarins (**4**) in excellent yields (82–97%) (Figure 2) [22].

## 3. Reactivity of 3-Mercapto-Coumarin

3-Mercapto-coumarin (**4**) contributed to the synthesis of many chain and fused compounds. Accordingly, the reaction of 3-mercapto-coumarin (**4**) with some acrylonitriles and acrylates under the Michael addition condition created *S*-acetonitrile **6a**, *S*-propanenitrile **6b**, *S*-ethayl acetate **6c**, and *S*-propanoate **6d** coumarin derivatives, respectively (Figure 3) [18]. In another publication on the Mannich reaction, 3-mercapto-coumarin (**4**) condensed with formaldehyde and produced 3-hydroxy-methylthio-coumarin (**7**). The latter reacted with diphenylamine to give the corresponding α-aminomethylated thioether (**8**) (Figure 3) [23].

In a Chinese patent (2016), the author disclosed a method to fabricate benzothiophene-2-carboxylic acid (**9**) via a phase-transfer catalyst of 3-mercapto-coumarin (**4**) under high-pressure, 0.8–1.2 MPa (Figure 4) [24].

*N*-Acetyl-S-(3-coumarinyl)cysteine (**11**), which could be isolated from rat urine [25], was synthesized by the reaction of 3-mercapto-coumarin (**3**) and *N*-acetyl-3-chloro-*D*,*L*-alanine methyl ester (**10**) (Figure 5) [26].

## 4. 4-Mercapto-Coumarin

The synthesis of 4-mercapto-coumarin by methods based on 4-hydroxycoumarin has already been discussed [19,27,28,29,30].

In 1970, Peinhardt and Reppel allowed 4-hydroxycoumarin (**12**) to react with phosphorus oxychloride to get 4-chlorocoumarin (**13**). The latter, under reaction with potassium hydrosulfide in situ prepared from potassium hydroxide with methanol saturated with hydrogen sulfide (H_2_S), gave the corresponding 4-mercapto-coumarin (**14**) in a good yield, 90% (Figure 6) [27].

Recently, the synthesis of 4-mercapto-coumarin (**14**) occupied the scope of interest of Ghosh’s work as an in situ transformed intermediate to synthesize different coumarin-fused heterocycles via 4-hydroxycoumarin (**12**) [19,28,29,30]. Dissolving the 4-hydroxycoumarin (**12**) in pyridine followed by the addition of toluene-4-sulfonyl chloride led to the formation of the tosyl derivative (**15**). Treatment of the latter with NaSH in ethanol furnished the corresponding 4-mercapto-coumarin (**14**), which succeeded by transformation to the final product (Figure 7).

## 5. Reactivity of 4-Mercapto-Coumarin

In 1975, Eiden and Zimmermannhe synthesized diphenylacetyl thioester (**16**) and biscoumarinyl sulfide (**17**) via the reaction of 4-mercapto-coumarin with 2,2-diphenylethen-1-one according to Figure 8 [31].

In the previous example of Ghosh’s work, 4-mercapto-coumarin served as a transitional compound to produce different coumarin-fused heterocycles employing 4-hydroxycoumarin (**12**) as a starting reactant. As the compound (**14**) was produced, it converted immediately to the final products (Figure 7).

Accordingly, various 2*H*-thiopyrano[3,2-*c*][1]benzopyran-5-ones (**19**) [28,29] and 4-aryloxymethylthiopyrano[3,2-*c*][1]benzopyran-5(2*H*)-ones (**21**) [19,30] were prepared through the thio-Claisen rearrangement of 4-propargylthio[1]benzopyran-2-ones (**18**) and 4-[4-aryloxybut-2-ynylthio][1]benzopyran-2-ones (**20**) (Figure 9). Compounds **18** and **20** were prepared based on a two-phase mixture of 4-mercapto-coumarin (**14**) with propargyl halides and 1-chloro-4-aryloxybut-2-yne, respectively (Figure 9).

Regioselective synthesis of coumarin-annulated sulfur heterocycles, *cis*-benzothiopyrano[3,2-*c*]benzopyran-7(2*H*)-ones (**24**), was reported through aryl radical cyclization. The corresponding 4-[(2-bromobenzyl)sulfanyl]-2*H*-chromen-2-ones (**23**) was in situ prepared from a reaction between 4-mercapto-coumarin (**14**) and tributyltin hydride (**22**) in the presence of a radical initiator (AIBN) (Figure 10) [32].

In another publication, some thieno[3,2-*c*][1]benzopyran-4-ones (**27**) were synthesized by thermal thio-Claisen rearrangement of 4-allylthio[1]benzo-pyran-2-ones (**26**) (Figure 11). Compounds **26** resulted from a basic catalyzed reaction between 4-mercapto-coumarin (**8**) and different allylic halides (**25**). Without being separated from the reaction medium, compounds **26a**–**d** ended in four different derivatives via phase-transfer-catalyzed alkylation using TBAB or BTEAC as a catalyst. The differentiation of the end products depended on the alkyl substitutions (R_1_, R_2_) on the allyl halide, which influenced the mechanism of the cyclization during the final step (Figure 11) [33].

Nematollahi et al. investigated the electrochemical oxidation of catechols (**28**) in the presence of 4-mercapto-coumarin (**14**) as the nucleophile in water/acetonitrile (50/50) solution. Through an EC mechanism and in a one-pot process, 4-(dihydroxyphenylthio)-2*H*-chromen-2-one derivatives **29a** and **29b** were afforded (Figure 12) [34]. In another work of the same group, they explored the reactivity of catechol (**28**) and 4-mercapto-coumarin (**14**) in the presence of potassium ferricyanide as an oxidizing agent (decker oxidation) to develop thieno[3,2-c]chromen-6-onederivatives (**30**) (Figure 12) [35].

A series of 3-chloro-1-(5-((2-oxo-2*H*-chromen-4-yl)thio)-4-phenyl thiazol-2-yl)-4-substituted phenyl azetidin-2-ones (**36**) were synthesized in five sequential steps with the participation of 4-mercapto-coumarin (**14**) in addition to acetophenone (**31**), thiourea, and chloroacetyl chloride [36] (Figure 13). The synthesized compounds showed potent antimicrobial activity against *Staphylococcus aureus*, *Escherichia coli*, *Pseudomonas aeruginosa*, *Streptococcus pyogenes*, *Aspergillus niger*, *Aspergillus clavatus*, and *Candida albicans* [36].

## 6. 6-Mercapto-Coumarin

The structure creation of 6-mercapto-coumarin was performed following the synthesis of 3-SH and 4-SH-coumarin. This 6-SH-coumarin is an unstable compound that reacts directly with halo-compounds to give S-alkyl coumarin derivatives.

In 1999, Majumdar and Biswas reported 6-mercapto-coumarin (**39**) as an unstable compound [37]. 6-Mercapto-coumarin (**39**) was generated in situ from the disulfide (**38**) (Figure 14). This reduction was achieved with zinc dust in acetic acid in the presence of 6 N sulfuric acid by heating at 80 °C until the solution became clear. 6-Mercapto-coumarin (**39**) was used without further purification for the synthesis of 6-(4-aryloxybut-2-ynylthio)[1]benzo-pyran-2-ones (**41**) by its reaction with l-aryloxy-4-chloro- but-2-ynes (**40**) (Figure 14) [37].

## 7. 7-Mercapto-Coumarin

To our knowledge, only one article discussed the synthesis of 7-mercapto-coumarin (**46**) [20]. Therein, 7-hydroxycoumarin (**42**) was treated with sodium hydride and subsequently reacted with dimethylthiocarbamoyl chloride (**43**) to yield the 2-oxo-2*H*-chromen-7-yl dimethylcarbamate (**44**). The latter was subjected to a Newman–Kwart-type rearrangement to form S-(2-oxo-2*H*-chromen-7-yl) dimethylcarbamothioate (**45**). Cleaving of the carbamate group of **45** afforded 7-mercapto-coumarin (**46**) [20] (Figure 15).

One of the 7-mercapto-coumarin derivatives with the widest applications is 7-mercapto-4-methyl coumarin (MMC). MMC has a wide range of applications in the field of biology and material science. This substrate is considered as a frontier to synthesize varieties of bioactive compounds [38,39]. It is also indicated as a suitable matrix for the analysis of small molecular compounds [40]. The other described incorporation of this highly valuable scaffold is as a reporter molecule [41], reporters for thiol interactions at the nanoparticle surface [42], fluorescent probe [43,44,45], sugar acceptor [46], Raman reporter [47,48,49,50,51,52,53,54,55,56], an excellent substrate for fluorescence spectroscopy [57], photodimerizable and healable reactant [58], fluorescent dye [59,60], and probe molecule on gold-coated silicon nanowires [61].

### 7.1. Synthesis of 7-Mercapto-4-Methyl Coumarin (MMC)

The general procedure for the synthesis of 7-mercapto-4-methyl coumarin (**50**, MMC) depends on 7-hydroxy-4-methyl coumarin (**47**). The reaction of 7-hydroxy-4-methyl coumarin (**47**) with dimethylthiocarbamoyl chloride, was followed by a thermos-rearrangement that afforded the corresponding *S*-4-methyl-2-oxo-2*H*-chromen-7-yl dimethylcarbamothioates (**49**) (Newman–Kwart-type rearrangement) (Figure 16). The latter, upon hydrolysis in the presence of NaOCH_3_/CH_3_OH and acidification by HCl, produced the target 7-mercapto-4-methyl coumarin (**50**, MMC) (Figure 16) [21,62,63,64,65,66].

### 7.2. Reactivity of 7-Mercapto-4-Methyl Coumarin (MMC)

#### 7.2.1. Utilization of 7-Mercapto-4-Methyl Coumarin in the Synthesis of Bioactive Compounds

Novel 7-mercapto-coumarin derivatives (**53**) were designed starting from 7-mercapto-4-methyl coumarin (**50**). Most of compounds in **53** exhibited strong α1 antagonistic activity [65]. In particular, compound **53c** showed excellent activity, which was better than that of the reference compound prazosin [65]. 7-((4-(4-(2-Methoxyphenyl)piperazin-1-yl)butyl) thio)-4-methyl-2*H*-chromen-2-one (**53c**) was synthesized via the reaction of 7-mercapto-4-methyl coumarin (**50**) with 1,2-dibromoethane to give **51**, which in turn reacted with 1-(2-methoxyphenyl)piperazine (**52c**) (Figure 17) [65].

In an effort to find inhibitors of the bacterial enzyme DNA gyrase, Miller and his co-workers developed a potential low-molecular-weight inhibitor of 7-[4-(4-tert-butyl-benzyloxy)-1*H*-indazol-3-ylmethylsulfanyl] 4-methylcoumarin (**57**) (Figure 18) [67]. The protected and brominated product (**56**) was involved in thioether formation with MMC (**50**) by nucleophilic substitution to give **56**. The phenolic protection of **56** was removed and formed ether linkage simultaneously with BnBr or *p*-*t*Bu-BnBr in presence of fluoride ion. The removal of the Boc protection was performed simply in acidic condition to give **57**. Compound **57** was 10 times more potent than the reference drug, novobiocin as a DNA gyrase inhibitor [67].

Lee and his co-workers synthesized analogs of DCK, which was known to be active against HIV. In these new structures (thia-DCK), the sulfur atom was the isosteric equivalent to the oxygen atom in the original structure. These derivatives (**61a** and **61b**) were synthesized by fusing the derivatized thiane ring within the benzene moiety of coumarin through a four-step reaction (Figure 19) [21,64]. They proved to be potent as an anti-HIV agent with an EC_50_ value of 0.14 and 0.039 µM and a remarkable therapeutic index of 1110 and 1000 for **61a** and **61b**, respectively [21,64].

Another 12 antiviral agents of *S*-substituted 7-mercapto-4-methyl coumarin analogs (**62**) were synthesized and evaluated against HBV in HepG2 cells (Figure 20) [62]. These series of derivatives were prepared from reaction between MMC and halo compounds assisted by K_2_CO_3_/KI. The IC_50_ of **62a** and **62b** as anti-HBsAg activities was (0.01 µmol/L), which was 16-fold more potent than the reference (3TC). Compounds **62c**–**f** exhibited interesting inhibitory activity toward both HBsAg and HBeAg [62]. Another approach belonging to Chen et al. was to prepare and evaluate the effectiveness of **63** as an antitumor agent [68]. This compound, which afforded by reduction of **62b**, showed a broad spectrum of activity against four tumor cells, as well as remarkably increased cellular apoptosis in a concentration-dependent manner. Furthermore, it induced A549 cell cycle arrest at the G2/M phase [68].

In 2014, Liu et al. aimed to prepare furoxan-based nitric oxide (NO) releasing S-coumarin, 4-(2-(4-methyl-2-oxo- 2H-chromen-7-ylthio)ethanoxy)-3-(phenylsulfonyl)-1,2,5-oxadiazole 2-oxide (**66**) (Figure 21). It was attained by the formation of an ethyl linker between MMC and the NO-releasing moiety (**65**). This linker was established by the reaction of chloroethanol with MMC in a K_2_CO_3_-containing solvent. The resulting intermediate (**64**) reacted with compound **65** in DCM, while DBU acted as a catalyst to deliver compound **66**. This latter showed antiproliferation activity on A549, HeLa, A2780, A2780/CDDP, and HUVEC cell lines with IC_50_ (µM) of 0.12, 0.024, 0.036, 0.14, 0.22, respectively [69].

Guo et al. synthesized 4-methyl-7-thiocyanato-2*H*-chromen-2-one (**67**), which may be used as an inhibitor of monoamine oxidase A or anti-influenza drug [70]. Cutting off the C–S bond through a photocatalysis of inorganic thiocyanates salt delivered the green “CN”, which transformed 7-SH group of 7-mercapto-4-methyl coumarin (**50**) in the presence of a 10 W white light and 1 mol% Rose Bengal to 7-SCN (**67**) (Figure 22) [70].

#### 7.2.2. 7-Mercapto-4-Methyl Coumarin as a Fluorophore Probe

Fluorescent probes based on 7-mercapto-4-methyl coumarin (MMC) are widely reported in the literature [71,72,73,74]. In contrary to 7-mercapto-4-methyl coumarin’s poor emission characteristic, its thiol-alkylated analog shows high fluorescence.

A specific fluorescent probe based on monosulfanyl-coumarin-BODIPY for the selective detection of cysteine in living cells and artificial urine has been synthesized via a simple substitution reaction on the 5-position of XDS-BOD-XDS:BODIPY-Cl_2_ (**68**) with the thiol group generated from 7-mercapto-4-methyl coumarin (**50**) to yield MC-BOD-XDS (**69**), XDS-BOD-XDS (**70**) (Figure 23) [75]. The reactivity could be attributed to the free SH-coumarin, which quickly binds to another MC-BOD-XDS and produces strong red fluorescent XDS-BOD-XDS (**70**) [75].

Özer et al. presented a new fluorescent chemosensor for some transition metals, which was obtained by conjugating two molecules of 7-mercapto-4-methyl coumarin (**50**) through a glyoxime bridge. This dimer was formed by a refluxing mixture of mercapto-coumarin, (*E*,*E*)-dichloroglyoxime DCGO, and NaHCO_3_ in MeOH. The coumarin collaborates in this conjugate by its fluorophore property while vic-dioxime acts as a metal-chelating moiety. This chelating capability was expressed in the last step of the following scheme to deliver **72** (Figure 24) [76].

Hili et al. used the 7-mercapto-4-methyl coumarin (**50**) as a fluorescent tag, to demonstrate the cyclic peptide conjugation strategy through the nucleophilic ring-opening of an aziridine moiety during the macrocyclization of linear peptides enabled by amphoteric molecules (Figure 25) [77].

Navarro et al. labeled the modified cellulose nanofibrils (CNFs) with furan and maleimide moieties by the fluorescent probe, 7-mercapto-4-methyl coumarin, through the thiol-Michael reaction (Figure 26) [78]. The fluorescein/coumarin labeled cellulose nano-fibrils (FC-CNFs) avoid a dye-to-dye interaction (for the same molecule) with an expected wide biological application such as multimodality molecular imaging [78].

Hajdu and his co-workers reported the synthesis and enzymological characterization of three fluorogenic phosphatidylcholine analogs PC-1 (**77**), PC-2 (**78**), and PC-3 (**79**), targeting the detection and the quantitative assays of phospholipase A2 (sPLA2) [79]. Each demonstrated molecule contained a 7-mercapto-4-methyl coumarin fluorophore and 2,4-dinitroaniline quencher on both tails (Figure 27) [79]. The small size of these molecules helps to not disrupt the natural membrane.

Gold(III)–thiolato complexes based on cyclometallated pyrazine-centered pincer ligands form a new class of photoluminescent gold compounds. The luminescence behavior of these gold-core compounds depends on the arrangement of supramolecule in the solid and liquid forms [80]. (C^N^pz^^C)AuSR (**81**) was prepared by the described method and isolated as a yellow to red solid (Figure 28) [80]. In this formula, C^N^pz^^C represents 2,6-bis(4-ButC_6_H_4_)pyrazine dianion and R is 7-mercapto-4-methyl coumarin [80].

Choi and co-workers developed a naphthalimide–coumarin conjugate (NC) typically through thioether linkage (Figure 29) [81]. 6-Bromo-2-butyl-benzo[de]isoquinoline-1,3-dione (**82**) reacted with 7-mercapto-4-methyl coumarin (**50**) in the presence of potassium carbonate to give 2-butyl-6-((4-methyl-2-oxo-2*H*-chromen-7-yl)thio)-1*H*-benzo-[de]isoquinoline-1,3(2*H*)-dione (**83**) (Figure 29A) [81]. The molecular flexibility and the aggregation of the nanoparticles of the investigated compound (**83**) enhanced the aggregation-induced emission (Figure 29B) [81].

#### 7.2.3. 7-Mercapto-4-Methyl Coumarin as Photodimerizable and Healable Reactant

Zhao et al. exploited the photodimerization character of 7-mercapto-coumarin to introduce a novel bio-sourced self-healing technique. The epoxidized cottonseed oil was used as the main reagent, and it was photocrosslinked in the presence of 0.25 equivalents of 7-mercapto-4-methyl coumarin (**50**) as a photodimerizable and healable reactant. The reaction was initiated with 2 wt.% of a photo-based generator porphobilinogen (PBG) (Figure 30) [82].

#### 7.2.4. 7-Mercapto-4-Methyl Coumarin for Thioglycosylation Reaction

The reaction of 7-mercapto-4-methyl coumarin with glycosides and their possible applications have been in the scope of many researchers [83,84]. One such group was Tanaka et al., who investigated the synthesis of 4-methyl coumarin-7-yl-*α*-*S*-glycosides 87 by Williamson condition starting from *N*-acetyl-neuraminic acid, *N*-glycolylneuraminic acid, or 3-deoxy-d-*glycero*-d-*galacto*-2-non-ulopyranosonic acid (KDN) (**85**) (Figure 31). The product is efficient as fluorogenic substrates for tracking and quantitative analysis of the bacterial enzyme neuramidase [85].

Enzymatic synthesis of *S*-glycosides (**89**) from the reaction of 7-mercapto-4-methyl coumarin (**50**) as aromatic thiol acceptors and 4-nitrophenyl-β-D-glucuronide (*p*NP-GlcA) (**88**) as a sugar donor (glucuronide donor) was reported (Figure 32) [86]. The reaction was efficiently glycosylated by *Dt*GlcA-E396Q (the mutated form of β-D-glucuronidase *Dt*GlcA) as a biocatalyst with a 51% yield in pKa 5.03 [86].

Yoshida et al. developed a novel oligosaccharide-labeling with, 7-mercapto-4-methyl coumarin (MMC) as the detachable fluorescent tag linked to the anomeric center of unprotected sugar (**92**) (Figure 33) [87]. The resulting MMC-labeled sugars (**92**) showed a high sensitivity for fluorescence detection and could be used for the quantification of oligosaccharide mixtures [87].

#### 7.2.5. 7-Mercapto-4-Methyl Coumarin as Metal Chelator (Complex Formation)

7-Mercapto-4-methyl coumarin (**50**) is characterized by its ability to accept transition metals and form metal chelates that play a prominent role in the development of coordination chemistry [88,89].

A diironhexacarbonyl cluster covalently linked to S-4-methyl coumarin (**93**) was synthesized (Figure 34) [90]. The complex, **93**, is electrochemically unstable and exhibited photoinduced intramolecular electron transfer from coumarin to the iron-carbonyl unit [90].

Rank et al. demonstrated a bis(coumarin thiolate) complex (**96**) to constitute an interesting building block for multimetal structures in a “complexes as ligands” approach [88]. The reaction of the terpyridine ligand, 4-*t*butyl-4′-(4-pyridinyl)- 2,2′-bipyridine (**94**), with [PtCl_2_(dmso)_2_] yielded the corresponding complex *cis*-[PtCl_2_(L)] (**95**). The coupling reaction of with 7-mercapto-4-methyl coumarin led to the formation of the bis(coumarin thiolate) complex [Pt(4-methyl-coumarin-7-thiolate)_2_(4-*t*butyl-4′-(4-pyridinyl)-2,2′-bipyridine)] (96) (Figure 35) [88].

Carboni et al. synthesized novel (C^N^N) cyclometalated Au^III^ complexes with the general formula [Au(bipydmb-H)X][PF_6_] (bipydmb-H is C^N^N cyclometalated 6-(1,1-dimethylbenzyl)-2,2′-bipyridine) (Figure 36) [91]. The [Au(bipydmb-H)(MeQS)][PF_6_] (3-PF_6_) including coumarin expressed distinctive biological activity as an anticancer agent against human lung epithelial cancer (A549) and human ovarian cancer (SKOV-3) cells [91].

Novel metal-free and metallophthalocyanines with 7-thioether-4-methyl coumarin were prepared from the reaction of 7-mercapto-4-methyl coumarin (**50**) with 1,2-dicyano-4-nitrobenzene (**99**) to give 7-(3,4-dicyanophenylthio)-4-methyl coumarin (100) (Figure 37) [92]. Cyclotetramerization of this structure under heating with dry 2-*N*,*N*-dimethylaminoethanol in a sealed tube afforded 2,9,16,23-tetrakis(7-coumarinthio-4-methyl)-phthalocyanine (**101a**). The latter chelated various metals in a reaction with their salts, Zn(CH_3_COO)_2_·2H_2_O, NiCl_2_·6H_2_O, CuCl, and CoCl_2_·6H_2_O, and gave the corresponding metallophthalocyanines **101b**–**e** (Figure 37) [92].

## 8. Conclusions

Coumarins are one of the heterocyclic structures of great interest in the development of valuable structures with both biological and industrial applications.

Mercapto (or sulfanyl)-coumarins represent an interesting class of compounds that open many possibilities for further chemical transformations. As a nucleophile, mercapto-coumarin can be used to prepare derivatives with halides, activated halides, and in nucleophilic aromatic substitutions with the appropriate aromatic halides. They can also be used in Michael addition and Mannish reactions. Additionally, they can serve as an intermediate for the synthesis of thiophenes fused to coumarin as well as for the preparation of a thiocoumarin or a thiochromone ring.

Remarkably, 7-mercapto-4-methyl coumarin (MMC) shows the most useful structure for different applications. This substrate is considered as a starting compound to synthesize varieties of bioactive compounds. In addition, it plays an important role in materials science, where it serves as a reporter molecule, fluorescent probe, sugar acceptor, metal chelator (complex formation), and Raman reporter.

Finally, 5 and 8 mercapto-coumarins leave a lot of space for further investigation as there are not described in the literature. Even 8-mercapto-4,6-dimethyl coumarin was claimed to have been obtained [93], no proof of this structure was given.

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
