# Peer review of "Insight on Mercapto-Coumarins: Synthesis and Reactivity"

_molecules, 2022, doi:10.3390/molecules27072150_

Round 1

Reviewer 1 Report

The present manuscript is devoted to an interesting topic. The authors have found a poorly explored class of heterocyclic derivatives and made a review on their properties. Despite high significance of such a review, in my opinion, it has several serious flaws, and thus, in the present form it should be totally revised.

  1. Krueger Chemische Berichte, 1923, vol. 56, p. 481 - describes 8-mercaptocoumarins. This paper should be briefly mentioned in the review.
  2. The introduction section should explain what time period is discussed in the review and why. It also should give information on earlier reviews on the topic or state their absence.
  3. According to the abstract, this review covers papers from 1987 to 2021. But there is no reviews on 4-mercaptocoumarins, 6-mercaptocoumarins and 7-mercaptocoumarins. There is only a review on 3-mercaptocoumarins (ref. 18). Thus, I think this review should expand the time period or give a strong explanation on ignoring time period before 1987 for 4-mercaptocoumarins, 6-mercaptocoumarins, 7-mercaptocoumarins and 8-mercaptocoumarins.

Author Response

Dear Professor

Editor, Molecules

Thank you for giving us the opportunity to resubmit our work (ID: molecules-1646737) to published in your respectable journal.

I am grateful to you and the reviewers for the valuable suggestions provided. I have tracked the comments, and I responses all the comments during the manuscript with highlighted yellow.

Herein the responses to the reviewer's comments:

Reviewer 1

Point 1:  Krueger Chemische Berichte, 1923, vol. 56, p. 481 - describes 8-mercaptocoumarins. This paper should be briefly mentioned in the review.

Response

The article considered the possibility of obtaining 8-thiol from coumarin that substituted at positions 4 and 7 with a methyl group. And there is no evidence of the chemical structure of the compounds obtained, the author just guess. We referred to this in the conclusion with the addition of reference 93.

Point 2: The introduction section should explain what time period is discussed in the review and why. It also should give information on earlier reviews on the topic or state their absence.

Response:

The topic has not been previously addressed or discussed (as a review article) since work began on the preparation of mercaptocoumarin compounds.

The mentioned period is related to what was made available to us by the scientific sites, Reaxy, Scifinder and Google Scholar. Accordingly, the period in our review article is related to the oldest research article until the most recent research article.

Point 3: According to the abstract, this review covers papers from 1987 to 2021. But there is no reviews on 4-mercaptocoumarins, 6-mercaptocoumarins and 7-mercaptocoumarins. There is only a review on 3-mercaptocoumarins (ref. 18). Thus, I think this review should expand the time period or give a strong explanation on ignoring time period before 1987 for 4-mercaptocoumarins, 6-mercaptocoumarins, 7-mercaptocoumarins and 8-mercaptocoumarins.

Response:

Thank you for pointing us to the important point that taking into account your vision

After a deep search, it was confirmed that these compounds weren't prepared in the time period before 1987 except for 4-mercaptocoumarin. The scientific sites have been confirmed the synthesis of 4-mercapto-coumarin from 4-chlorocoumarin, as we supported the manuscript with an article from Pharmazie journal in 1970.

Reviewer 2 Report

This work by El-Sawi and colleagues describes the synthesis and reactivity of mercapto-coumarins.

I consider the manuscript suitable for publication once the authors address the following points in any subsequent revision:

  1. They should incorporate some future perspective of this research field, offering their opinion as experts in the field. 
  2. Furthermore, a small introductory paragraph on each compounds in the corresponding sections should be incorporated in the main text.
  3. The final conclusions must be comprehensive, not just a summary.

Author Response

Dear Professor

Editor, Molecules

Thank you for giving us the opportunity to resubmit our work (ID: molecules-1646737) to published in your respectable journal.

I am grateful to you and the reviewers for the valuable suggestions provided. I have tracked the comments, and I responses all the comments during the manuscript with highlighted yellow.

Herein the responses to the reviewer's comments:

Point 1: They should incorporate some future perspective of this research field, offering their opinion as experts in the field.

Response: Thanks for your suggestion. Checked and added

Point 2: Furthermore, a small introductory paragraph on each compounds in the corresponding sections should be incorporated in the main text.

Response: Thanks for your proposal. Checked and added

Point 3: The final conclusions must be comprehensive, not just a summary.

Response: Thanks for your recommendation. Checked and added

Round 2

Reviewer 1 Report

The authors have made all necessary changes. Now the manuscript can be accepted.

Reviewer 2 Report

I consider the manuscript suitable for its publication once the English has been revised and some typos (for example, page 16-line 481, word "compoundto": a space is missing) have been corrected.